# Role of IL-17 in Morphogenesis and Dissemination of *Cryptococcus neoformans* during Murine Infection

**DOI:** 10.3390/microorganisms10020373

**Published:** 2022-02-05

**Authors:** Nuria Trevijano-Contador, Elena Roselletti, Rocío García-Rodas, Anna Vecchiarelli, Óscar Zaragoza

**Affiliations:** 1Mycology Reference Laboratory, National Centre for Microbiology, Instituto de Salud Carlos III, Carretera Majadahonda-Pozuelo, 28222 Madrid, Spain; rocio.rodas@gmail.com; 2Department of Experimental Medicine, Microbiology Section, University of Perugia, 06123 Perugia, Italy; e.roselletti@exeter.ac.uk (E.R.); vecchiar10@gmail.com (A.V.)

**Keywords:** *Cryptococcus neoformans*, C57BL76J, *Il17a^−/−^*, IL17, Titan cells

## Abstract

*Cryptococcus neoformans* is a pathogenic yeast that can form Titan cells in the lungs, which are fungal cells of abnormally large size. The factors that regulate Titan cell formation in vivo are still unknown, although an increased proportion of these fungal cells of infected mice correlates with induction of Th2-type responses. Here, we focused on the role played by the cytokine IL-17 in the formation of cryptococcal Titan cells using *Il17a^−/−^* knockout mice. We found that after 9 days of infection, there was a lower proportion of Titan cells in *Il17a^−/−^* mice compared to the fungal cells found in wild-type animals. Dissemination to the brain occurred earlier in *Il17a^−/−^* mice, which correlated with the lower proportion of Titan cells in the lungs. Furthermore, knockout-infected mice increased brain size more than WT mice. We also determined the profile of cytokines accumulated in the brain, and we found significant differences between both mouse strains. We found that in *Il17a^−/−^*, there was a modest increase in the concentrations of the Th1 cytokine TNF-α. To validate if the increase in this cytokine had any role in cryptococcal morphogenesis, we injected wild-type mice with TNF-α t and observed that fungal cell size was significantly reduced in mice treated with this cytokine. Our results suggest a compensatory production of cytokines in *Il17a^−/−^* mice that influences both cryptococcal morphology and dissemination.

## 1. Introduction

*Cryptococcus neoformans* is a pathogenic yeast that causes significant morbidity and mortality in immunosuppressed people, including those with HIV/AIDS, solid organ transplants, or other immune impairment [1,2,3]. This yeast is of particular interest because it has several well-characterized mechanisms that allow its survival in the host and the escape from the immune response.

The capsule is its main phenotypic feature and the best-characterized virulence factor. It confers physicochemical characteristics but also plays a very important role in the interaction with the host. The capsule is essential for the survival of *C. neoformans* inside macrophages [4] and also protects it from free radicals and other attacks on the immune system [5]. In addition, *C. neoformans* undergoes a typical morphological change, which involves a significant increase in both cell and capsule size, resulting in the appearance of cells of an enlarged size denominated as Titan cells [6,7,8] that can reach a size of 80–100 microns in vivo [6,7,9,10]. Recently, several groups described the in vitro conditions that mimic this transition [11,12,13]. These have unraveled some of the factors that induce this process, such as serum, CO_2_, and hypoxia [11,12,13,14,15]. However, there are still important aspects about the formation of these cells in vivo that remain uncharacterized. In particular, little is known about the function that Titan cells have during infection. Titan cells contribute to the persistence of the yeasts in the lungs since they cannot be phagocytosed or eliminated. In addition, they inhibit phagocytosis of other cells of regular size and induce a nonprotective Th2-type immune response [16].

An intriguing aspect of Titan cell formation is its relationship with host factors. Early characterization of this phenomenon indicated that the proportion of Titan cells in mouse lungs was particularly high in asymptomatic infection models, which could indicate that these cells participate in the latency phase [7]. In addition, there is evidence that the host immune response also plays an important role in determining the morphology of cryptococcal cells in the lungs. In *rag1^−/−^* knockout mice (which do not produce IgM antibodies), *C. neoformans* cells have a larger diameter compared to the yeast cells found in wild-type mice [17].

In a previous work, we showed that the proportion of Titan cells also depends on the mouse strain used in the infection model. This proportion is particularly high in C57BL/6J compared to CD1 mice. This difference correlates with the different immunological responses elicited by these two types of animals; thus, the formation of Titan cells is associated with a decreased production of IFN-γ, TNF-α, and IL-17 and an increase in Th2-type responses [16]. In this sense, it has also been shown that Th1 responses depending on TNF-α and Th17 confer protection during cryptococcosis [18,19].

The Th17-type response is promoted by the cytokines IL-17, IL-6, and IL-23. As with Th1-type cytokines, IL-17, IL-22, and IL-23 cytokines have been associated with protection against infection [18,19]. In addition, there are studies that show that protection against *C. neoformans* is associated with an increase in the production of antimicrobial peptides (AMPs) [20]. IL-17A and IL-22 induce the production of AMPs and acute-phase proteins such as β-defensins, S100A8, S100A9, lipocalin-2, and serum amyloid protein A3 (SAA3) in bronchial epithelial cells [21,22,23,24,25]. S100A8 and S100A9 form a heterodimer called calprotectin [26,27,28] that is involved in neutrophil recruitment [21,23,24] and has fungistatic activity against *C. neoformans* [29,30,31].

The proinflammatory cytokine TNF-α has been associated with the elimination of the microorganism and greater protection [19,32]. In contrast, a type of Th2 response has been associated with a lack of protection against this infection [33].

In this work, we investigated the role of IL-17 in Titan cell formation. Paradoxically, we found that in *Il17a^−/−^* in KO mice, there was a lower proportion of Titan cells compared to WT mice, which also correlated with an increase in the production of TNF-α in KO mice. Our results support new functions of Th1 and Th17 responses in the control and development of this infection.

## 2. Materials and Methods

### 2.1. Yeast Strain and Growth Conditions

*Cryptococcus neoformans* H99 strain was used throughout the work [34]. Yeasts were incubated in Sabouraud liquid medium (Oxoid Ltd., Basingstoke, Hampshire, England) at 30 °C with shaking (150 r.p.m.). For solid media, we added 1.5% agar to the media.

### 2.2. Mouse Strains

Six- to eight-week-old male mice from C57BL/6J (Charles River Laboratories) and knockout IL-17A (*Il17a^−/−^)* on C57BL/6J genetic background (generated at the Center for Experimental Medicine and Systems Biology, University of Tokyo, Minato-Ku, Tokyo, Japan) were used. The animals were kept in ventilated racks at 22–24 °C with proper environmental enrichment (cupboard houses and hollow cylinders). All animal experiments were performed in agreement with the EU Directive 2010/63 and the National Law 116/92. The protocol was approved by the Perugia University Ethics Committee (Permit Number: 223/2016-PR). All animals were housed in the animal facility of the University of Perugia (Authorization Number: 34/2003A). In some experiments, animals were housed at the Instituto de Salud Carlos III animal facility. In that case, procedures were approved by the Bioethical Committee and Animal Welfare of the Instituto de Salud Carlos III (CBA2014_PA51) and of the Comunidad de Madrid (PROEX 330/14) and followed the current Spanish legislation (Real Decreto 53/2013).

### 2.3. Infections

Cells grown in liquid Sabouraud as described above overnight were washed with PBS and suspended at 3.3 × 10^7^ cells/mL in the same buffer. Thirty microliters of this suspension (10^6^ cells) was introduced intranasally in each mouse that was anesthetized with a subcutaneous injection of a mixture of tiletamine/zolazepam-xylazine (50-5 mg/kg). Animal health was monitored by evaluating body weight loss, evasion instinct, ruffled hair, and hunched body. Based on these parameters, a score from 1 to 6 was given to each animal. Mice were sacrificed after 6 and 9 days of infection unless they reached a clinical score of >6 or suffered a body weight loss above 25%. In this case, the mice were sacrificed for humanitarian reasons. Animals were sacrificed by placing them in a chamber that contained a high CO_2_-enriched atmosphere. The experiments were performed 3 times on different days.

### 2.4. Administration of TNF-α

TNF-α (GenScript, 1 mg) (200 µg/Kg) or PBS (same volume) was injected intraperitoneally 1 h prior to the cryptococcal infection. As described above, mice were then anesthetized and infected intranasally (i.n.) with 10^6^ cells per mouse. Mice were sacrificed following the above-described criteria 48 h after infection.

### 2.5. Determination of CFUs and Fungal Morphology

Lungs and brains were excised and placed within cell strainers (100 µm size pore, BD Falcon, Louisville, CO, USA) with 10 mL of sterile PBS with collagenase A (0.5 mg/mL, Roche, Switzerland). The organs were homogenized using the plunger of a 5 mL syringe. After this, the samples were diluted in PBS (1/10, 1/100, and 1/1000), and 100 µL was plated on Sabouraud plates. The plates were incubated at 30 °C for 2 days, and we then counted the number of colonies in each plate. In this way, we calculated the total number of living yeasts in each organ. The size of the yeast cells was obtained by mixing 4 µL of the organ extracts with India Ink (Remel Bactidrop, Thermo Scientific, Waltham, MA, USA) on glass slides. The samples were observed using a Zeiss Axio Observer Z1 equipped with Apotome and a digital camera Axiocam MRm (Zeiss, Oberkochen, Germany). In each cell, we measured the total cell diameter (which included the capsule) and the cell size (cell wall) using Adobe Photoshop 7.0 (San Jose, CA, USA). Capsule size was calculated as the difference between these two parameters. To avoid any bias, different fields of the slide were imaged by two different people, and the images obtained were measured independently. The size of 50–200 cells from each mouse was measured.

### 2.6. Macroscopic Analysis of Brain and Lungs

The weight of the lungs, brain, and spleen from sacrificed mice was obtained, and we calculated the relative weight of each organ with respect to the total weight of the mouse.

### 2.7. Histology

To analyze the microscopic structure of the lungs, a small portion was taken and fixed in formalin (10%). The samples were then dehydrated and embedded in paraffin following the standard protocols of the Histology Service of the University of Perugia. Sections of 3.5 µm were obtained from each sample, placed on slides, and stained with hematoxylin/eosin. Finally, the samples were observed with a Zeiss Axio Observer Z1 equipped with Apotome and a digital camera (Zeiss, Oberkochen, Germany).

### 2.8. Identification of Different Types of Immune Cells by Flow Cytometry

To identify different types of immune cells, lungs were homogenized in 2 mL of PBS using 100 μm filters (BD Falcon, Louisville, CO, USA). They were then centrifuged at 3000 g in an Allegra ™ 6R centrifuge (Beckman Coulter, Brea, CA, USA). The pellets were suspended in 1 mL of PBS, and suspensions were made at 10^6^ cells/mL in flow cytometer tubes. Cells were washed and fixed in 1.5% p-formaldehyde for 10 min. The samples were then centrifuged and suspended in PBS. For cell labeling, the following mouse antibodies were used: anti-CD3e-PE (0.5 µg) to label T-lymphocytes, anti-Ly-6G-FITC (GR-1, clone 1A8-Ly6g, 0.5 µg) for polymorphonuclear (PMN) cells (Thermo Fisher Scientific, Waltham, MA, USA), and anti-CD14-PE (1 µg) for monocytes and macrophages (BD, Biosciences, San Jose, CA, USA). Staining was performed for 10^6^ cells/mL by using each antibody individually in single tubes for each animal in triplicate. Autofluorescence was checked by using unlabeled cells, whereby binding specificity was determined by measuring isotype control samples using anti-mouse IgG-PE and anti-mouse IgG-FITC (Sigma-Aldrich, Saint Louis, MO, USA). Labeling was performed for 20 min at room temperature and in the dark. Samples were washed using FB buffer (PBS + FBS) before measuring them on a BD FACSCalibur flow cytometer using CellQuest software version 3.3. The same software was also used for data analysis. Briefly, live cells were identified by excluding the debris on a 2D scatter plot (FSC-H vs. SSC-H). Aggregates were excluded by analyzing the scatter pulse signals (FSC-A vs. FSC-H and SSC-A vs. SSC-H). Afterwards single cells were analyzed on histograms for the individual fluorescence intensity of PE or FITC, representing CD3e+, Gr1+, and CD14+ cells. Quantification of T-lymphocytes, PMN, and monocytes/macrophages was performed by analyzing the percentage, evaluating the absolute cell count from total lung cells, and normalizing the cell counts per lung.

### 2.9. Measurement of Cytokine Concentration in Lungs and Brains

The organs were homogenized in 2 mL of PBS, and the homogenate was centrifuged at 3000 g in an Allegra™ 6R centrifuge (Beckman Coulter, Brea, CA, USA). For response type Th1, the cytokines TNF-α, IFN-γ, IL-1β, IL-12, and IL-6 were measured; for type Th2, IL-4, and IL-10 were measured; and for type Th17, IL-17, IL-21, IL-22, and IL-23 were measured. For this, ELISA Ready-SET-Go kits (eBioscience Inc., San Diego, CA, USA) were used. Cytokines were expressed as pg/mL of lung extract.

### 2.10. Statistics

To determine the statistical test to apply, we first evaluated the normality of the samples with the Kolmogorov–Smirnov test (non-normal distribution when *p* < 0.1). For normally distributed samples, we applied one- or two-way ANOVA and Student *t*-tests. When a nonparametric distribution was obtained, we used the Kruskal–Wallis and Mann–Whitney tests. *p* values were obtained using GraphPad Prism 5 (GraphPad Software. Inc, San Diego, CA, USA). For survival analysis, we applied the log-rank test, and significant differences were considered when *p* < 0.05.

## 3. Results

### 3.1. C. neoformans Formed a Lower Proportion of Titan Cells in Il17a^−/−^ Mouse Model

First, we focused on the role played by the cytokine IL-17 in the formation of this type of cell, and to do this, we used *il17a^−/−^* KO mice, which are deficient in this cytokine. Infections were carried out in wild-type mice (WT) and KO *Il17a^−/−^* mice with the H99 strain. Groups of five animals per condition were sacrificed after 6 and 9 days of infection. After this time, the morphology of the yeasts in the lungs was analyzed (Figure 1, top panels). As shown in Figure 1, Titan cells were observed in both mouse strains at 9 days of infection. However, in the wild-type mouse strain, the mean cell size of *C. neoformans* was around 50 microns, while in *Il17a^−/−^* mice, there was a lower proportion of these cells, and their mean size was between 35 and 40 microns (*p* < 0.0001). When analyzing the total cell size, we found that the proportion of Titan cells (>30 µm) was around 95% in wild-type mice, while in *Il17a^−/−^* mice, it was 83% (Figure 2A). This correlated with differences in cell body size, and we found that in wild-type mice, the percentage was around 60% compared to 35% in IL-17-deficient mice (Figure 2B).

### 3.2. C. neoformans Caused Increased Inflammation in Brains of Il17a^−/−^ Mice

When we calculated the organosomatic index, we observed that the lungs increased significantly in size after infection in both mouse strains compared to the lungs of uninfected animals, although we found no difference between WT and KO mice (Figure 3). With respect to the brain, in both cases, there was also an increase in size in infected mice, although it was more marked in KO mice. In the case of the spleen, and in contrast to the brain, the wild strain C57BL/6J significantly increased the size of this organ in infected mice compared to the *Il17a^−/−^* strain (Figure 3B).

### 3.3. Correlation of C. neoformans Morphology with Dissemination to Central Nervous System

Next, we studied whether there was a correlation between *C. neoformans* morphology and CNS dissemination. For this, after 6 and 9 days of infection, the lungs and brains of the mice were extracted, homogenized in PBS, and the number of CFUs was determined. As shown in Figure 4A,B, no significant differences in fungal burden in the lungs were observed between both types of animals. On the other hand, the number of CFUs in the brain was significantly higher in *Il17a^−/−^* mice compared to wild mice after 6 days of infection (Figure 4C), while this difference was not observed at 9 days (Figure 4D). These results suggest that in mice deficient for IL-17, the dissemination of *C. neoformans* to the CNS occurs earlier than in wild-type C57BL/6J mice.

### 3.4. Analysis of Cytokine Production during Infection

As shown in Figure 5, differences in the concentration of cytokines were observed in the lungs of both mouse strains. Regarding Th1-type cytokines, although not statistically significant, we found that there was a trend to find higher concentrations of TNF-α in *Il17a^−/−^* mice compared to wild-type animals (Figure 5A). IFN-γ increased in both WT and KO mice after infection, although this increase was stronger in WT mice. No differences were obtained when we measured other Th1 cytokines. IL-4 and IL-10 (Th2-type cytokines) increased in response to *C. neoformans* in the same way in both mouse strains, confirming that C57BL/6J mice have a Th2-type polarized response. Finally, in Th17 response type cytokines, there was a significant increase in IL-21 during infection in C57BL/6J mice that was not observed in *Il17a^−/−^* mice. IL-22 was similarly increased in wild animals and *Il17a^−/−^*. Finally, IL-17 was increased in infected wild-type mice, and as expected, it was not detected in *Il17a^−/−^* mice.

We also measured the concentration of these cytokines in the brain after 9 days of infection. As shown in Figure 6, the production of all cytokines tested increased in the brains of *Il17a^−/−^* mice. The concentration of the cytokines TNF-α and IFN-γ (Th1) was higher in the infected animals (*p* < 0.05) compared to the controls (without infection), and this increase was much more marked in the *Il17a^−/−^* mice than in wild C57BL/6J mice (infected). IL-10 (Th2) concentration was also higher in the brains of infected *Il17a^−/−^* mice compared to controls (*p* < 0.05). Regarding the Th17 cytokines, all increased in the *Il17a^−/−^* mice than in the wild-type mice, but only IL-21 increased significantly compared to the control (*p* < 0.05). The cytokines IL-1β, IL-6, and IL-4 did not accumulate in the brains of either of the two mouse strains (data not shown).

### 3.5. Cell Recruitment in Lungs

To quantify cell recruitment in the lungs, we measured the number of lymphocytes, macrophages, and polymorphonuclear cells (PMNs) by flow cytometry (see Materials and Methods (Section 2)) in mice from both mouse strains after 9 days of infection. As shown in Figure 7, in infected C57BL/6J mice, the proportion of lymphocytes, macrophages, and PMNs increased after infection, which was not observed in infected *Il17a^−/−^* mice.

### 3.6. Exogenous Administration of TNF-α Reduced Proportion of Titan Cells in Lungs

The decrease in the proportion of Titan cells in infected mice correlated with an increase in the proinflammatory cytokine TNF-α. Although the increase in the lungs was not statistically significant, the trend observed was in agreement with previous findings of our group, where we observed that in mice that developed a Th1-polarized response, there is also a significantly lower proportion of Titan cells in the lungs [16]. To confirm whether TNF-α had a direct inhibitory role in cryptococcal Titan cell formation, we administered this cytokine to C57BL/6J mice and examined its effect on cryptococcal morphology in the lungs. We observed that the administration of this cytokine resulted in a clear reduction in the proportion of Titan cells in the lungs and in the diameter of fungal cells (*p* < 0.0001) (Figure 8).

## 4. Discussion

*Cryptococcus neoformans* is characterized as yeast that changes in size during infection. This increase can occur by enlarging the capsule or both the capsule and cell body size. As a result, *C. neoformans* can form Titan cells during in vivo. These changes make the population of yeast found in the lungs during infection very heterogeneous, which contributes to evading the immune response. To date, little is known about the factors and molecular mechanisms responsible for the morphogenesis of *C. neoformans* in vivo. In general, Titan cell formation is a complex process and requires not only the activation of signaling pathways of the yeast but also specific inducing factors from the host [7,10,35,36,37].

In a previous work of our group, we investigated how the host environment influences the formation of Titan cells by comparing cryptococcal morphology in different mouse strains, CD1 and C57BL/6. We described that the proportion of Titan cells was much higher in C57BL/6J than in CD1, and this difference correlated with a different immune response polarization, so the lower percentage of Titan cells in CD1 correlated with the induction of Th1 responses. We hypothesized that under a nonaggressive environment for yeasts in the lungs (Th2), the yeasts significantly increase in size and form more Titan cells [16]. Furthermore, a possible relationship between the cytokine IL-17 with prevention in the dissemination of *C. neoformans* to the CNS has already been suggested by previous work [18]. Despite the experimental differences between our work and this article (mouse strains, inoculum, infection times, use of KO mice vs. depletion of IL-17 with Abs, etc.), our results are in agreement with those previously published [18].

In this work, we carried out several approaches to delve into the factors of the immune system that regulate the formation of Titan cells, which were based on the modulation of the immune system and the use of knockout mice. First, we investigated the role of IL-17 in the morphogenesis of *C. neoformans* using mice that do not produce this cytokine [38]. IL-17 primarily stimulates macrophages and endothelial cells to produce factors that contribute to local inflammation and has been described as a key mediator during host defense against fungal infections [39,40,41,42,43]. When comparing the morphology of the yeasts recovered from the lungs of C57BL/6J and *Il17a^−/−^* mice, we observed that the average cell size of the yeasts was lower in the mice that do not produce IL-17. These results were initially unexpected since *Il17a^−/−^* mice are deficient in inflammatory response but confirmed that the absence of IL-17 has an effect on the morphology of *C. neoformans*.

We argued that the effect of IL-17 on cryptococcal morphology was related to a different immune response, so we next investigated if the absence of IL-17 had any effect on immune polarization. We found that cryptococcal disease induced inflammation in the lungs of both mouse strains, which was more noticeable in the brains of *Il17a^−/−^* mice. The lower proportion of Titan cells in the lungs of *Il17a^−/−^* mice correlated with higher dissemination to the brain [18]. Previous studies have shown that IL-17A is crucial in preventing the dissemination of *C. albicans* [44]. Other studies have also linked IL-23 in defense against chronic infection by *C. neoformans* [45]. IL-23 has been shown to enhance IL-17 production [46].

Furthermore, mice lacking IL-23 or IL-17 show a decreased cellular immune response [38,47]. Overall, our results support the hypothesis that the cytokine IL-17 is important in preventing yeasts from escaping from the lungs and spreading to the brain.

To understand the mechanism by which the cytokine IL-17 regulates yeast growth and dissemination in the mice, we also measured the profile of accumulated cytokines in the brain and lungs [42]. This approach confirmed that C57BL/6J mice have a Th2-type response. Interestingly, we found a modest increase in the concentration of TNF-α in *Il17a^−/−^* mice that was not observed in wild-type mice. Since C57BL/6J mice have a Th2-polarized response in response to *C. neoformans* and reduced production of TNF-α compared to other mouse models [16], we believe that this increase in *Il17a^−/−^* mice might reflect a compensatory mechanism to overcome the absence of IL-17, which is consistent with the fact that the yeasts in the lungs of these mice are smaller in size. In our previous work, an increase in the proportion of Titan cells in C57BL/6J mice compared to CD1 was also associated with a lower concentration of IFN-γ and TNFα. To test this hypothesis, we injected recombinant TNF-α in C57BL/6J mice and found that this significantly reduced the cryptococcal size in the lungs, confirming that TNF-α helps control and diminish the number of Titan cells. These data support our previous hypothesis that enhancement of Th1 responses creates a more aggressive environment for the yeast, making the process of Titan cell induction difficult. TNF-α is involved in the expression of some antimicrobial factors, such as reactive nitrogen species through iNOS expression activation [48]. TNF-α also induces the expression of NADPH oxidases and consequently the production of reactive oxygen species [49]. For this reason, we hypothesized that among others conditions, that increase the concentration of TNF-α result in a more stressful environment for the yeasts due to higher amount of ROS and RNS. Further experiments are required to investigate in detail the factors that regulate the induction of Titan cells in vivo.

This work highlighted several aspects that regulate the formation of Titan cells in mouse models. This morphological transition plays a key role in the adaptation of *C. neoformans* to the host. Our work contributes to unraveling how some key elements of the immune response, such as IL-17 and TNF-α, regulate this transition in vivo. This work can also help to understand how different patients with a different immunological state can have different susceptibility to develop cryptococcosis by having a different predisposition to accumulate more Titan cells in vivo.

## Figures and Tables

**Figure 1 microorganisms-10-00373-f001:**
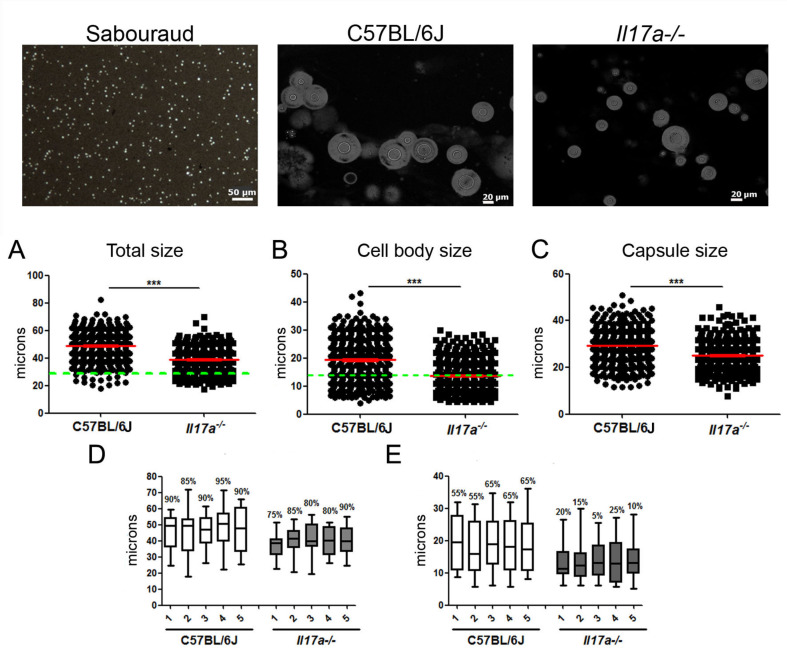
Morphology of *C. neoformans* in C57BL/6J and *Il17a^−/−^* mice (top panels). Cells of *C. neoformans* H99 (10^6^ cells/mouse) were inoculated male C57BL/6J and *Il17a^−/−^* mice as indicated in M&M. After 9 days of infection, lungs were isolated and homogenized. Size of yeasts was visualized after suspending extracts in India ink. (**A**) Distribution of total cell size, (**B**) cell body size, and (**C**) capsule size of *C. neoformans*. Red lines represent mean and standard error. Green lines indicate extent to which Titan cells were considered. Box and whisker plot of total cell size distribution. Line inside box represents median, and top and bottom lines represent 75th and 25th percentiles, respectively. Numbers above each distribution indicate percentage of Titan cells found in each animal. (**D**) Total cell size and (**E**) cell body size obtained in 5 mice of each strain. Asterisks indicate significant differences.

**Figure 2 microorganisms-10-00373-f002:**
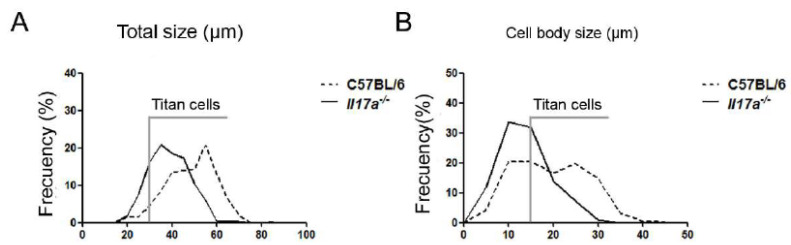
Size distribution of *C. neoformans* in C57BL/6J and *Il17a^−/−^* mice. Frequency representation of total cell size (**A**) and cell body size (**B**) of *C. neoformans* WT H99 cells isolated from C57BL/6J mice (dotted line) and *Il17a^−/−^* mice (black line) after 9 days of infection with a dose of 10^6^ cells/mouse.

**Figure 3 microorganisms-10-00373-f003:**
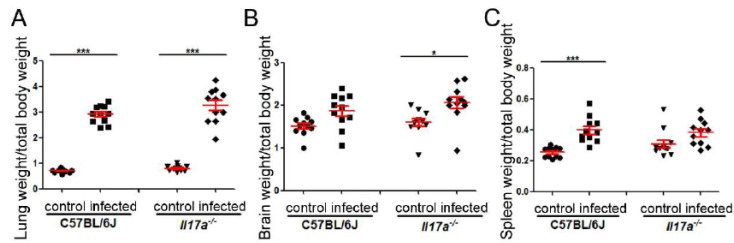
Effect of *C. neoformans* on target organ size in C57BL/6J and *Il17a^−/−^* animals. Weight of lungs (A), brains (B), and spleens (C) relative to total weight of mouse in C57BL/6J and *Il17a^−/−^* mice after 9 days of infection with H99 strain of *C. neoformans*. Asterisks indicate significant differences.

**Figure 4 microorganisms-10-00373-f004:**
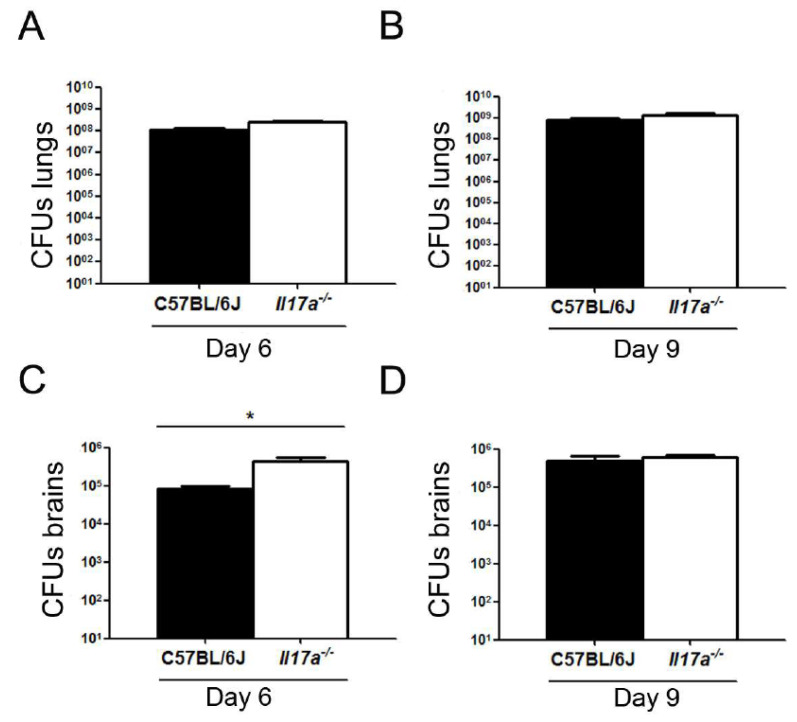
Fungal burden and spread of *C. neoformans* in C57BL/6J and *Il17a^−/−^* mice. C7BL/6J mice (black bars) and *Il17a^−/−^* (white bars) infected as described in M&M were sacrificed after 6 (**A**,**C**) and 9 (**B**,**D**) days of infection, and number of CFUs was quantified in lungs (**A,B**) and brains (**C,D**). Asterisk indicates statistically significant differences.

**Figure 5 microorganisms-10-00373-f005:**
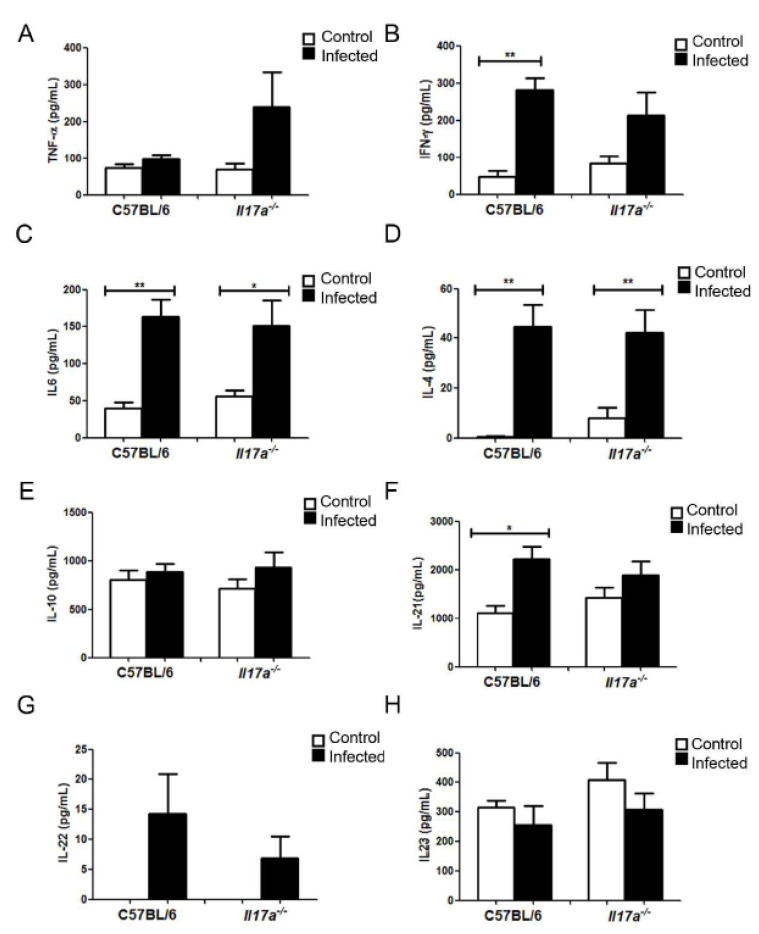
Cytokine concentration in lungs of C57BL/6J and *Il17a^−/−^* mice. C57BL/6J and *Il17a^−/−^* mice were infected with 10^6^ yeast strain H99 per mouse (black bars) or treated with PBS (white bars). Concentration of TNF-α (**A**), IFN-γ (**B**), IL-6 (**C**), IL-4 (**D**), IL-10 (**E**), IL-12 (**F**), IL-22 (**G**), and IL-23 (**H**) was determined from lung extracts after 9 days of infection, as indicated in M&M. Figure represents mean and standard deviation of five different mice. Asterisks indicate significant difference between control and infected mice.

**Figure 6 microorganisms-10-00373-f006:**
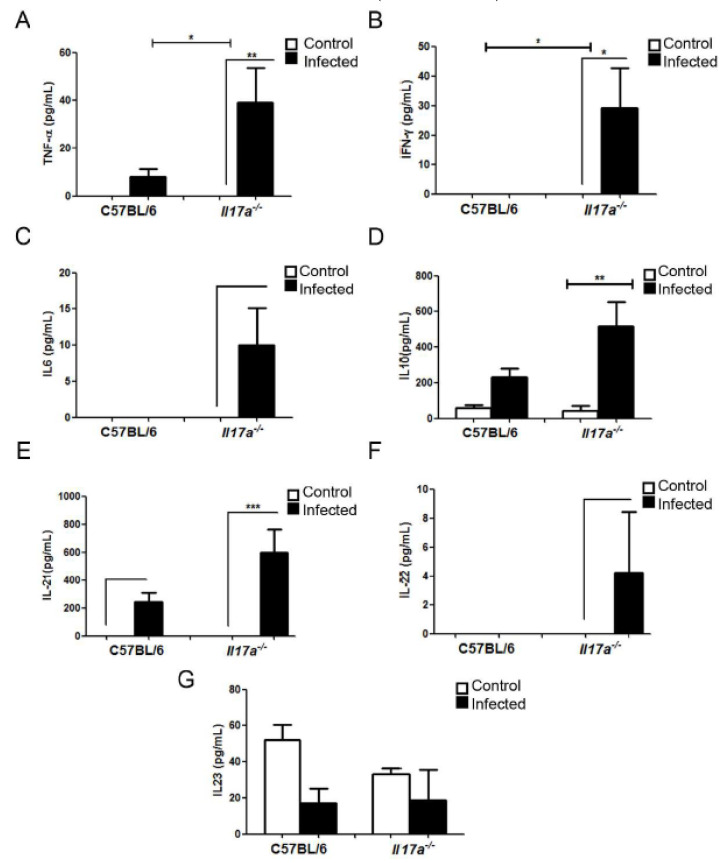
Cytokine concentration in brains of C57BL/6J and *Il17a^−/−^* mice. Mice were infected as indicated in Figure 7, and concentration of TNF-α (**A**), IFN-γ (**B**), IL-6 (**C**), IL-10 (**D**), IL-21 (**E**), IL-22 (**F**), and IL-23 (**G**) was determined from brain extracts after 9 days of infection. Figure represents mean and standard deviation of five different mice. Asterisks indicate significant difference between control and infected mice.

**Figure 7 microorganisms-10-00373-f007:**
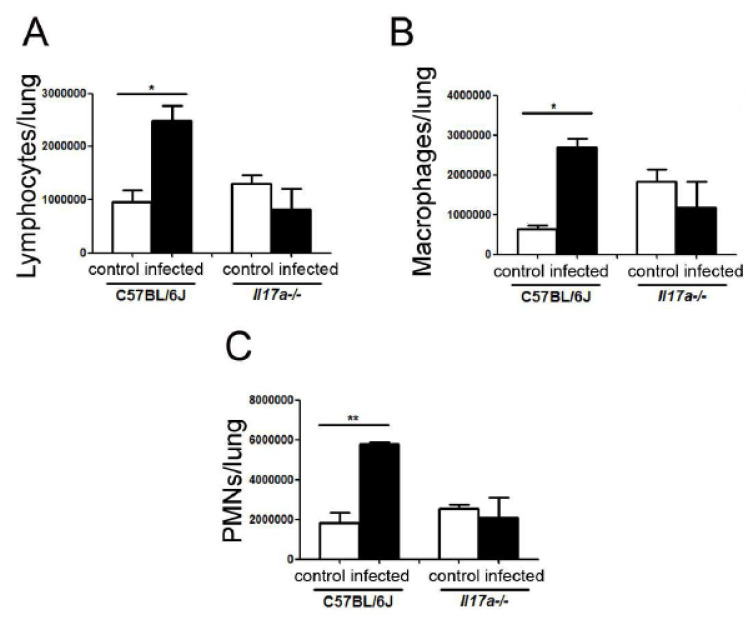
Recruitment of immune cells in lungs of C57BL/6J and *Il17a^−/−^* mice. Animals were infected as described in M&M, and after 9 days, lungs (**A**–**C**) were removed. Organs were homogenized as described in M&M, and total number of lymphocytes (**A**), macrophages (**B**), and PMNs (**C**) was quantified. Asterisks indicate significant differences.

**Figure 8 microorganisms-10-00373-f008:**
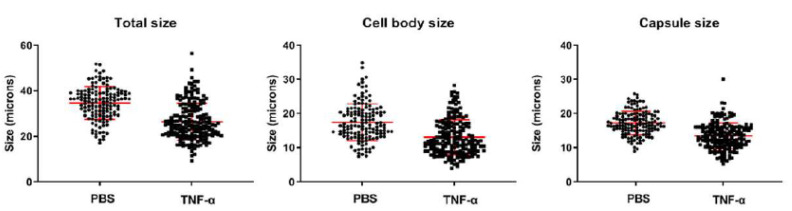
Size of *C. neoformans* in lungs of mice after administration of TNF-α. (**A**) Distribution of total cell size, (**B**) cell body size, and (**C**) capsule size of *C. neoformans* H99 cells isolated from lungs of C57BL/6J after 48 h of infection with a dose of 10^6^ cells/mouse. Groups of 4 mice were administered TNF-α 200 µg/Kg or PBS (same volume) intraperitoneally 1 h prior to cryptococcal infection. Experiment was repeated twice with similar results.

## Data Availability

Not applicable.

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
