# Peer review of "Role of IL-17 in Morphogenesis and Dissemination of *Cryptococcus neoformans* during Murine Infection"

_microorganisms, 2022, doi:10.3390/microorganisms10020373_

Round 1

Reviewer 1 Report

The authors addressed  all of my comments. 

Reviewer 2 Report

Dr. Joanna Uszko,

Assistant Editor Microorganisms

Dear Dr. Uszko,

The Manuscript ID: microorganisms-1572529 entitled “Role of IL-17 in morphogenesis and dissemination of Cryptococcus neoformans during murine infection” was well improved and the few typo mistakes fixed. I really enjoyed reading the manuscript and in this reviewer opinion the manuscript is now suitable to be published in Microorganisms.

Specific comments:

Please, standardize the use of the word “Figure” in the whole manuscript. In some paragraphs it is written with the first letter in capital and in other not.  

This manuscript is a resubmission of an earlier submission. The following is a list of the peer review reports and author responses from that submission.

Round 1

Reviewer 1 Report

The authors investigate the role played by the cytokine IL-17 in the formation of cryptococcal Titan cells using IL17a knockout mice. They found a lower proportion of Titan cells in the IL-17KO mice at day 9 post infection compared to wild type mice, increased dissemination in the brain of the KO mice and some small changes in cytokine levels between the KO mice and the wild type. The authors over inflate the finding that there were higher but not significant changes in the TNF levels in the lungs of the IL-17 KO mice.   The link between increased cryptococcal dissemination to the brain in IL-17 KO mice has already been shown (PMID:21359196).  Previous work from this group has already shown a link between TH1 cytokines and the formation of Titan cell, so it was not surprising that when they treated mice with a TH1 cytokine there was a difference in the number of Titan cells. The link between reduced Titan cell formation in the IL-17 KO mice and the change in TNF is tangential at best.

The number of mice used, and the number of experiments done are not clear. It looks like the data throughout the paper is from only one experiment with only 5 mice.

Was proteinase inhibitor added to the supernatants used for the cytokine measurement? This was not included in the M&M section. If not, cytokines can degrade in the freezer which could change the results.   

In figure 6, while there are trends that suggest that there may be difference in TNF, there was not a significant difference between the strains of mice.

On line 267 the authors state that “but only IL-21 and IL-22 increased significantly” this is not what the figure shows, only IL-21 was significantly increased not IL-22.

Authors need to show strategy for flow. GR1 is not a good antibody for PMNs as it stains both Ly6C and Ly6G cells, PMNS are only Ly6G+.

The authors only show the PMN from the brain but not the other cell types, was there any difference in the lymphocytes or macrophages? Additionally, it looks like the cells from the brain were isolated the same way as the lung cells. If so, this is not the appropriate way to isolate the cell from the brain for flow.

Figures 1, 2, and 3 should be condensed into one figure as they all come from the same data. Would suggest that 2d and 2e be moved to the supplement.  

There is a typo in the abstract on line #24 “we infected TNF-a to wild type..”

Frequency is misspelled on the y-axis on both figure 3A and 3B

Typo on figure 5 line 237 “as described in figure 5..”

Author Response

ANSWERS TO REVIEWER 1

The authors investigate the role played by the cytokine IL-17 in the formation of cryptococcal Titan cells using IL17a knockout mice. They found a lower proportion of Titan cells in the IL-17 KO mice at day 9 post infection compared to wild type mice, increased dissemination in the brain of the KO mice and some small changes in cytokine levels between the KO mice and the wild type. The authors over inflate the finding that there were higher but not significant changes in the TNF levels in the lungs of the IL-17 KO mice.   The link between increased cryptococcal dissemination to the brain in IL-17 KO mice has already been shown (PMID: 21359196).  Previous work from this group has already shown a link between TH1 cytokines and the formation of Titan cell, so it was not surprising that when they treated mice with a TH1 cytokine there was a difference in the number of Titan cells. The link between reduced Titan cell formation in the IL-17 KO mice and the change in TNF is tangential at best.

Response: First of all, we would like to thank the reviewer for agreeing to read the manuscript as well as for all his/her comments and observations.

In reference to the paper named by the reviewer (PMID: 21359196), we would like to clarify that this is a study mainly focused on the role of IL17 in lung infection by C. neoformans. In addition, the approach used is very different from ours since they work with a different mouse strain (BALB/c). In addition, they do not use Knockout mice for IL-17 but rather they perform a partial depletion of this cytokine using anti-IL17 antibodies. Furthermore, the inoculum used and the infection times are different from those used by us. They cause infections of 10^4 cells mouse and during 35 days. In our case, to make sure we observed dissemination to the CNS we used a higher inoculum (10^6 cell/ mouse). In our conditions, IL17 knockout mice (where it was found that they did not produce any IL17) could not prolong the infection for more than 9 days. In the article mentioned by the reviewer, the authors measured fungal burden in the lung, spleen and brain after 28 days, and they suggest that IL17 could be involved in the prevention of dissemination to the brain. Our findings are in agreement with this idea. However, our work is mainly focused on the role of this cytokine (IL17) on the formation of Titan cells.

Based on these differences, and to address the reviewer´s comment, we have referenced this article in the discussion ,made modifications to the text (lines 324-328).

On the other hand, we would like to comment that the main objective of our research group is to elucidate the role of the Titan cells in both, virulence and in the response of the host to this phenomenon (see previous work of our group). Precisely, in the work referenced by the reviewer, (PMID: 26243235), we characterized two different mouse strains, each with a different type of immune response; CD1 (Th1-type response) and C57BL/6J (Th2-type response). We observed differences in Titan cells depending on the immune response. The results of this work encouraged us to investigate in depth how the Th17-type response would influence the formation of Titan cells.

The reviewer also states that we had previously shown a link between TH1 cytokines and the formation of Titan cell, so it was not surprising that when they treated mice with a TH1 cytokine there was a difference in the number of Titan cells. Although true, we respectfully disagree with the reviewer in the importance of the finding of this article. We previously found an association that lead to the hypothesis that TNF-α is important to control cryptococcal cell size in vivo. But, in the present work, we have demonstrated a cause-effect phenomenon, which confirms the direct role of TNF-α in the process of titan cell formation in C. neoformans. We believe that this experiment is sound and contributes to a better understanding of the factors that regulate cryptococcal size and Titan cell formation in vivo. For this reason, we also respectfully disagree with the statement that the findings are tangential at best. Although this work is an extension of our previous article in Cellular Microbiology, it provides new information and direct confirmation of the role of some immune mediators, such as TNF-α in cryptococcal morphogenesis.

Comment 1: The number of mice used, and the number of experiments done are not clear. It looks like the data throughout the paper is from only one experiment with only 5 mice.

Response 1: We agree with the reviewer that this information is not clear in the text. And although in some sections we have indicated the number of mice in the experiment (line 187 or line 208), it is true that the number of times that the animal infections have been carried out is not indicated. We did these experiments three times on different days with five animals per group. We have added this information in M&M (lines 115-116).

Comment 2: Was proteinase inhibitor added to the supernatants used for the cytokine measurement? This was not included in the M&M section. If not, cytokines can degrade in the freezer which could change the results.

Response 2:  The proteinase inhibitor wasn’t added to the supernatants because the cytokines quantification was evaluated immediately.

Comment 3: In figure 6, while there are trends that suggest that there may be difference in TNF, there was not a significant difference between the strains of mice.

Response 3: The reviewer is right, and we may have overstated our conclusions. However, the trend is clear, and this is consistent with our previous findings, where we observed that a decrease in the percentage of titan cells in the lung correlates with an increase of TNF-a. Furthermore, in the brain this difference was statistically significant. Despite the difference observed in our article is not significant, we believe that this trend is reflecting a different immune response of the C57BL/6J mice. It is noteworthy to mention that in our model, these mice show a Th2 polarized immune response, so any change in Th1 cytokines is of interest. We did not observe any other trend in any other cytokine. Finally, we have confirmed the role of TNF-a by injecting this cytokine directly in the mice, observing a significant decrease in size. In summary, we believe that despite the lack of significance of the graph, our data supports that TNF-a plays a role in controlling cryptococcal cell size during infection.

Following the reviewer´s comment, we have rewritten the manuscript to be more precise about the difference observed.

Comment 4: On line 267 the authors state that “but only IL-21 and IL-22 increased significantly” this is not what the figure shows, only IL-21 was significantly increased not IL-22.

Response 4: We regret this error in the text, the reviewer is right and as shown in figure 6 only IL-21 presents significant differences. This has been corrected (line 267).

Comment 5: Authors need to show strategy for flow. GR1 is not a good antibody for PMNs as it stains both Ly6C and Ly6G cells, PMNS are only Ly6G+.

Response 5: We apologize for the mistake. We did a mistake during the writing of the M&M section (line 155 “Anti-mouse Ly-6R (Gr1) (FITC) (0.5 mg / mL) for polymorphonuclear cells (eBioscience, Inc, San Diego, CA)”).

The antibody used was the Anti-mouse Ly-6G (Gr1) (FITC) of eBioscience. The sentence has been re-written in the text. We used the same antibody in previous manuscripts, for example PMID: 2603712. This error has been corrected in lines 154-155.

Comment 6: The authors only show the PMN from the brain but not the other cell types, was there any difference in the lymphocytes or macrophages? Additionally, it looks like the cells from the brain were isolated the same way as the lung cells. If so, this is not the appropriate way to isolate the cell from the brain for flow.

Response 6: We agree with the referee, different ways are also used to isolate the immune cells from the brain for the Flow Cytometry analysis. The mechanic dissociation/homogenization of the brain is acceptable and an established method (for example, PMID 24510618). We know that this technique is becoming outdated, but at the time these experiments were done in the laboratory, there was no other, more up-to-date protocol. We do not use Percol after homogenization as in the attached paper because the Percol is used only to purify the cells population and run in the FC only the leucocytes.

As we agree with the reviewer that there are better and more updated protocols to isolate brain cells. The fact that we did not detect the other cell types in the brain supports deleting this graph from the manuscript.

Comment 7: Figures 1, 2, and 3 should be condensed into one figure as they all come from the same data. Would suggest that 2d and 2e be moved to the supplement.

Response 7:  We appreciate the reviewer's suggestion but we believe that putting 3 figures together in one is a lot of information and is more confusing for the reader. We would like to keep panels D and E from Figure 2 because there you can see the percentage of Titan cells in each animal individually.

But following the reviewer´s recommendation, we have decided to put together figures 1 and 2, which are now figure 1

Comment 8: There is a typo in the abstract on line #24 “we infected TNF-a to wild type..”

Response 8: We appreciate this reviewer's observation and we regret the error in the abstract in line 24. We have modified the text and now it appears as " we infected wild type mice with TNF-α” (line 24).

Comment 9: Frequency is misspelled on the y-axis on both figure 3A and 3B

Response 9: We regret this error in figure 3, which has been modified and now appears as "Frequency" in figure 3A and 3B. Now Figure 2.

Comment 10: Typo on figure 5 line 237 “as described in figure 5.”

Response 10: We appreciate this reviewer's observation and we regret the error in the text of figure 5, line 237, which has been modified and is now indicated as "as described in Material & Methods"

Reviewer 2 Report

The article is interesting, the experiences are well planned and described. The only comment concerns the final announcement that the research will be continued on the current model. I strongly suggests the authors to consider the possibility of conducting research in an in vitro model, e.g. on cell cultures, instead of animal. However I understand that it is not always possible. And regarding Figure 1. Please, provide  all photos with the same magnification (photo from Sabouraud culture is different than from lungs)

Author Response

The article is interesting, the experiences are well planned and described. The only comment concerns the final announcement that the research will be continued on the current model. I strongly suggests the authors to consider the possibility of conducting research in an in vitro model, e.g. on cell cultures, instead of animal. However I understand that it is not always possible. And regarding Figure 1. Please, provide, all photos with the same magnification (photo from Sabouraud culture is different than from lungs).

Response: We thank the reviewer for the nice assessment of the work. We fully agree with the reviewer that in vitro models should be used whenever possible. In fact, our group published in 2018 in the journal Plos Pathogen along with other groups (see PMIDs 29775477, 29775480, 297754749) an in vitro medium for obtaining Titan cells. Since this discovery, we have always tried to use this in vitro model whenever the objectives of the project have allowed it (PMID 31988178). But in this case, the host's immune response, as well as the role of IL17, was important to see the morphogenesis and dissemination of Cryptococcus and that is why this project has been carried out in in vivo models. We have in fact planned to use our in vitro conditions to examine the direct role of some cytokines. However, using a cytokine outside of the in vivo context will likely fail to produce any effect, since cytokines have pleiotropic roles and require specific ligands to induce their effects.

On the other hand, we hope that the reviewer does not mind that we did not change the scale bars in figure 1. The reason they are different is that the Titan cells looked very disproportionate. We believe that as long as the scale of each photo is indicated it is not a problem.

Reviewer 3 Report

Dr. Joanna Uszko,

Assistant Editor Microorganisms

Dear Dr. Uszko,

The Manuscript ID: microorganisms-1478544 entitled “Role of IL-17 in morphogenesis and dissemination of Cryptococcus neoformans during murine infection” is a well written manuscript describing the role of TNF-alfa as an important cytokine to control the C. neoformans Titan cells formation in knockout mice of IL-17. In this reviewer opinion, the manuscript is scientifically sound and is of interest of Microorganisms’ public. Find bellow few suggestions in order to improve it.

Specific comments:

Line 20: Please, replace “Knockout” by “knockout”.

Line 64: There are two dots at the end of the sentence “…. type responses. [16].”

Line 147: Please, check the typo in “equipedwith”.

Line 172-173: Please replace the “e” by “and” in “IL-12 e IL-6”, “IL-4 e IL-10”, and “IL-22 e IL-23”.

Line 240: Please, delete “it” in the figure legend (….and the number of CFUs it was quantified in lungs…).

Line: 245: Please replace “significantly” by “significant”.

Line: 292: What about the description in the legend of the figure 7D (PMNs/brain).

Round 2

Reviewer 1 Report

While the authors did attempt to address some of my comments, the lack of care when revising the paper is concerning. The authors combined the original figure 1 and 2 but failed to write a new figure legend or provide the new figure. The authors say that they removed the brain flow data form figure7 but it is still there.

In response 2, the authors state that “proteinase inhibitor wasn’t added to the supernatants because the cytokines quantification was evaluated immediately.” This however is not what is written in the M&M section (See lines 170-171).  

What clone of the GR-1 antibody was used? This is important as one clone is specific for PMNs and one clone will bind to other cells as well as PMNs. Therefore, I asked for the gating strategy, which was not provided in the response.  

The section on the brain flow should be removed completely. The authors in their response state “The fact that we did not detect the other cell types in the brain supports deleting this graph from the manuscript.” The CD14 marker the authors use is also found on microglia and therefore the authors should have seen these cells in the flow if it was done properly.